# Application of Improved Asynchronous Advantage Actor Critic Reinforcement Learning Model on Anomaly Detection

**DOI:** 10.3390/e23030274

**Published:** 2021-02-25

**Authors:** Kun Zhou, Wenyong Wang, Teng Hu, Kai Deng

**Affiliations:** 1School of Computer Science and Engineering, University of Electronic Science and Technology of China, Chengdu 611731, China; huteng@caep.cn; 2Institute for Computer Application, China Academy of Engineering Physics, Mianyang 621900, China; dengkai@caep.cn

**Keywords:** reinforcement learning, asynchronous advantage actor-critic, anomaly detection, generative adversarial network

## Abstract

Anomaly detection research was conducted traditionally using mathematical and statistical methods. This topic has been widely applied in many fields. Recently reinforcement learning has achieved exceptional successes in many areas such as the AlphaGo chess playing and video gaming etc. However, there were scarce researches applying reinforcement learning to the field of anomaly detection. This paper therefore aimed at proposing an adaptable asynchronous advantage actor-critic model of reinforcement learning to this field. The performances were evaluated and compared among classical machine learning and the generative adversarial model with variants. Basic principles of the related models were introduced firstly. Then problem definitions, modelling processes and testing were detailed. The proposed model differentiated the sequence and image from other anomalies by proposing appropriate neural networks of attention mechanism and convolutional network for the two kinds of anomalies, respectively. Finally, performances with classical models using public benchmark datasets (NSL-KDD, AWID and CICIDS-2017, DoHBrw-2020) were evaluated and compared. Experiments confirmed the effectiveness of the proposed model with the results indicating higher rewards and lower loss rates on the datasets during training and testing. The metrics of precision, recall rate and F1 score were higher than or at least comparable to the state-of-the-art models. We concluded the proposed model could outperform or at least achieve comparable results with the existing anomaly detection models.

## 1. Introduction

Anomalies, also called outliers, exceptions or peculiarities are patterns in the data that do not conform to the expected behavior. Types of anomalies include point, contextual and collective ones which are classified based on the single data, context and relationships among collection of data, respectively. Anomaly detection [1] is the identification of rare items, events or observations which raise suspicions by differing significantly from the majority of the data. It has been proven critical to many applications such as network intrusion of recognizing potential cyber-attacks, credit card fraud, medical applications where electrocardiography or other biological data are monitored to detect the patients’ situation and video surveillance of identifying the suspicious movements.

### 1.1. Traditional Methods for Anomaly Detection

Anomaly detection research was once conducted by the statisticians using mathematical methods. Numerous algorithms such as extreme value, probabilistic and statistics, linear, spectral and proximity based were proposed in the past and achieved good performances. The common point of these methods is to quantify the deviations from the normal patterns for data points with a numerical score. The likelihood fit of data points to the model is the outlier score in probabilistic modeling, or density value for proximity-based modeling. The residual distance of data points to a lower dimensional data representation is the outlier score in linear modeling. In temporal modeling, a function of the distance from previous data points (or deviation from forecasted value) is used to create the outlier score. Typical models as listed in Table 1 include one-class Support Vector Machine (SVM), Isolation Forest (IF) and Local Outlier factor (LOF).

**Table 1 entropy-23-00274-t001:** Anomaly detection classification.

Classification	Representative Model	Type
Classifier based	One-Class SVM [2], unsupervised NN	Unsupervised Machine Learning
Nearest neighbors	KNN [3], LOF [4], COF [5], …	
Clustering based	CBLOF [5], LDCOF [5], …	
Statistical based	HBOS [5], …	
Subspace based	rPCA [5], …	
Ensembles and Combination	Isolation Forest [6], FeatureBagging [7]	Ensembles
Neural Networks	Auto-encoder with FNN, LSTM [8], …	Deep Neural Network
Generative model	GAN [9], …	
Others	Information theory based	Others
	Spectral decomposition based	
	Visualization based	
	Reinforcement Learning	

Although these methods were effective in detecting the anomalies, they also suffered from oversimplifying assumptions about data representations and poor algorithms scalability. They make different assumptions about the ’normal’ behavior which depend highly on data patterns of particular domains. Anomaly detection tasks become more challenging because of the insufficient knowledge and representative of the anomalies for specific systems. They faced the challenges of data labelling or baseline ’normal’ requiring domain knowledge and the property of time-evolving makes detection even harder. Furthermore, different types of data exhibit different abnormal behavior. For example, sequential anomalies are objects or instances over time steps which are obviously different to image or video surveillance. Large amount of data due to the pervasiveness of data-collecting devices are often unlabeled which pose computational challenges and thus require novel and efficient approaches.

In supervised learning there are theories and experimental evidences trained with stochastic gradient descend that local minima problems might be resolved. Although unsupervised, semi-supervised and supervised learning achieved certain successes in the anomaly detection field, the weaknesses of the three are also blatant. Unsupervised learning with no pre-classified labels assumed most of the data is normal while outliers or smaller clusters are anomalies. High-dimensional data is typically sparse there could be many small clusters which are not necessarily outliers. Although semi-supervised labels for normal data allows accurate modelling of normal behavior, false alarm rate due to unseen but legitimate instances are still high. Supervised with labels for normal data and known anomalies cannot detect unknown and emerging anomalies.

### 1.2. Deep Reinforcement Learning for Anomaly Detection

Recent years have seen great progresses in the development of AI especially the Deep Learning (DL) and Reinforcement Learning (RL) especially when Alphago by Deepmind using RL beat the human champion. Thereafter the application of RL in diverse domains emerged. Introducing DL and RL into anomaly detection and taking advantages of the two great powers to achieve even better performances has become the trend. Inspired by the successful application of RL to many tasks, we proposed the model for anomaly detection based on A3C reinforcement learning (A3C) with adaptable deep neural network. A3C learns policy and state value functions simultaneously and the reward function incentivizes the agent to collect as much reward as possible. For sequential anomalies attention based NN is proposed as the value function estimator of the actor and CNN based models for video/images tasks. DNN is employed to represent a reward function to detect whether a new observation follows a normal pattern. The objective is to establish the appropriate RL models to capture the objects or instances with different properties that deviate from normal behavior. Our proposed method can adapt to different domain tasks for better performances.

RL is different from supervised and unsupervised learning setup in that (1) the function are typically not independent identical distribution, (2) agents affect the data they will receive and (3) the feedback can be sparse and are always delayed. The modelling process and samples for training and testing of RL are different to supervised learning where labeled datasets are available. Further, the data distributions are often non-stationary which depend on how agents taking actions. The agents basically decide the data they want see which enhance the complexities of training.

Local minima problems persist for RL algorithms when the policy converges early to some deterministic points. One possible solution is to introduce the exploration and exploitation which is realized in our model using the entropy method. Deep RL methods face challenges of solving large control and prediction problems where states may exceed $10^{170}$ for the chess GO and even larger for infinite continuous state space. Value function evolved from lookup tables where states and state-action pair have entries to NN. There are too many states and/or actions to store in memory and too slow to learn the value of each state individually, also the states are often not fully observable. The proposed asynchronous global and workers model attempted to enhance the efficiency by utilizing the power of parallel computing.

## 2. Related Researches

We surveyed three types of papers, namely review, general purpose RL and RL applied to anomaly detection. Comprehensive surveys on network anomaly detection including algorithms, experiments and analyses were done in [10,11,12,13]. Deep Learning for anomaly detection were surveyed in [14,15]. Researches were on sequential anomaly detection using RL [16,17].

States, actions and rewards based on actions were defined using RL algorithms such as dueling DQN, A3C with the goals to optimally assign resources in [18,19,20]. To find falsifying inputs for CPS, states, actions and rewards were defined based on inputs, outputs and function of past-dependent output signal, respectively, using Double DQN and A3C algorithms [21]. Authors proposed PMU-RL method [22] to balance energy power consumption and efficiency which showed promising results for heterogeneous computing platforms. We summarized papers [23,24,25,26] as that states were defined according to the systems status, agents took certain actions and rewards were fed back to the agents. The goals are to maximizing the rewards to get better performances using RL related algorithms such as DQN, A3C, TRPO. To detect malicious websites (URLs), states are defined as vector space representation of website features such as HTTPS protocols, IP address, prefix or suffix in URLs, actions are 0 or 1 (0 for benign, 1 phishing URL) [27]. The goal is to get the most rewards based on action using DQN algorithm. Resource optimization of smart grids, motor anomalies and biological data with RL was addressed in [28,29,30], respectively. Network intrusion detection based on RL were proposed in [31,32,33,34]. Although all these paper claimed effectiveness of their general purpose RL models, more independent replays and extensive comparisons among SOTA model could be more persuasive.

Methods or frameworks were proposed for anomaly detection in [35,36,37]. Auto-encoders were used in the anomaly detections which achieved promising performances [38,39,40]. More extensive comparisons with SOTA models could be done to futher support the efficacies of their models. Anomaly detection based mainly on LSTM models were proposed in [41,42,43,44]. Researches suggest LSTM may be replaced by attention models so these authors should consider better modelling using attention such as transformers. Generative methods such as GAN were proposed to anomalies detection in [45,46]. Emphatic approach to the problem of off-policy temporal-difference learning was given in [47]. Comparison with exisiting models could be done extensively especially for novel approaches. Authors proposed methods and tested on the dataset of CICIDS with the precision of 99.1% in [48]. Weighted Extreme Learning Machine (ELM) method was proposed for the intrusion detection and the results showed precision of around 99% in [49,50]. More benchmark datasets tests could be done and more metrics should be used to corroborate their conclucions. The A2C were proposed for the sequential anomaly detection task in [51,52]. DQN algorithm suffers from overestimations which was not previously known was confirmed in [53]. Their adaptation method reduced the overestimation situation and improved the performances. As we know A3C might improve the performaces compared to A2C, so the paper should provide more evidences that their models based on A2C were superior to A3C.

All authors claimed test results of their proposed approaches outperformed existing models in their respective domains. There are at least two problems which were not addressed satisfactorily by these solutions. The NN structures of RL framework (especially the NNs for state values and the actions) were not clear or just employing the original fully connected NNs. Furthermore, comparison of application performances using A3C based on RL with GAN models were not sufficient. This paper therefore aimed at proposing appropriate NN structures of RL for specific domains and compared the performances of state of the art A3C of RL with the GAN in anomaly detection.

Contributions of this paper were summarized as follows:1We elaborated multiple related DL and RL methods’ basic principles, modelling process and made a comparative introduction on the strength and weakness of individual methods.2We proposed the improved RL architecture of A3C framework using appropriate policy and value DNNs for specific domains. For sequential tasks the RNN structure with attention mechanism for state value and CNN for images were proposed.3Performances of the proposed anomaly detection models were compared with machine learning models, GAN and with RL model variants. The empirical tests on three benchmark datasets showed that our proposed method outperformed or at least was comparable to the SOTA models in detecting anomalies.

In Section 1, we briefly introduced the background information. Related researches were reviewed in Section 2. Methodologies were presented in Section 3. We tested our models using experiments in Section 4. In Section 5, we presented conclusions and discussed possible future developments.

## 3. Preliminaries

### 3.1. Reinforcement Learning

We briefly introduced some key concepts of agent, environment, state, policy, reward and state/action value function.

The key success factors of RL applied to the anomaly detection are the problems definitions (the environments, states, actions and rewards), appropriate modeling architectures and the training samples for the respective domains. In RL agents sense the environment in discrete time steps and map the inputs to state information. RL agents execute actions and observe the feedback from the environment in the form of positive or negative rewards. Agents first act based on states and receive a reward, then observe state changes in the environment and update the policy to optimize the potential reward (see Figure 1 for the interaction between agent and environment). The optimal policy’s goal is to find the one that maximizes the rewards obtained over the time.

Agents could be classified as value based without policy, policy without value function, and actor-critic with both policy and value function. State of the art agents include Deep Deterministic Policy Gradient (DDPG), DQN with memory replay and target network, A2C, A3C. Policies define the behavior of agents and can be viewed as mapping function from states to actions. The policy is modeled as p(a|s) meaning the probability of agents taking action a under given state s. Policies could be realized in NN as the states and actions are the input and output, respectively. The NN models map from states/observations and actions to future states/observations. The state-value function of an MDP (in Equation (Equation 1) [54]) is the expected return starting from state s and then following policy π ($G_{t}$ is the total reward from state s on to the end of episode, γ is the decaying factor):(1)$$\begin{array}{cc} v_{\pi }\left(s\right) & =E_{\pi }\left[G_{t}|S_{t}=s\right]=E_{\pi }\left[R_{t+1}+\gamma v_{\pi }\left(S_{t+1}\right)|S_{t}=s\right]\\ \; & =\sum_{a}\pi \left(a|s\right)\sum_{s^{\prime },r}p\left(s^{\prime },r|s,a\right)\left[r+\gamma v_{\pi }\left(s^{\prime }\right)\right] \end{array}$$

MDP under fixed policies becomes MRP which can be solved using matrix computation. Fixed means the reward is the function of pair (state, action) but not the probability distribution. The value functions are expressed as in Equation (Equation 2) [54]. Given an MDP M = (S,A,P,R,γ) and a policy π, the state sequence $S_{1},S_{2}$, … is a Markov process (S,$p^{\pi }$) and the state reward sequence $S_{1},R_{2},S_{2}$, … is a MRP (S, $p^{\pi }$, $R^{\pi }$, γ).
(2)$$\begin{array}{cc} v_{\pi }\left(s\right) & =\sum_{a\in A}\pi \left(a|s\right)\left\{r\left(s,a\right)+\gamma \sum_{s^{\prime }\in S}T\left(s^{\prime }|s,a\right)v_{\pi }\left(s^{\prime }\right)\right\}\\ \; & =r_{s}^{\pi }+\gamma \sum_{s^{\prime }\in S}T_{s^{\prime }s}^{\pi }v_{\pi }\left(s^{\prime }\right) \end{array}$$
where $r_{\pi }^{s}$ stands for reward at state s of the agent π, $T_{s^{\prime }s}^{\pi }$ is the state transition matrix from state *s* to *s*’ under policy π. The value function can be expressed using matrix form $\vec{v_{\pi }}\;=r^{\pi }+\gamma T^{\pi }\vec{v_{\pi }}$, where $\vec{v_{\pi }}$ is a vector and $T^{\pi }$ stands for the transition matrix. The iterative method is used to solve the matrix. The State-action value function $q_{\pi }\left(s,a\right)$ (Equation (Equation 3) [54]) is the expected return starting from state *s*, taking action *a*, and then following policy π.
(3)$$\begin{array}{cc} q_{\pi }\left(s,a\right) & =E_{\pi }\left[G_{t}|S_{t}=s,A_{t}=a\right]\\ \; & =E_{\pi }\left[R_{t+1}+\gamma q_{\pi }S_{t+1},A_{t+1}\right)\left|S_{t}=s,A_{t}=a\right] \end{array}$$

Figure 2 illustrated the computation process of $v_{\pi }\left(s\right)$ and $q_{\pi }\left(s,a\right)$. The expectation is often simplified as the random reward in practice and then get Equation (Equation 4) [54].
(4)$$q_{\pi }\left(s,a\right)=\sum_{s^{\prime },r}p\left(s^{\prime },r|s,a\right)\left[r+\gamma \sum_{a^{\prime }}\pi \left(a|s\right)q_{\pi }\left(s^{\prime },a^{\prime }\right)\right]$$

Policy gradient is employed to minimize the loss of policy using stochastic gradient and is expressed in Equation (Equation 5) [54].
(5)$$▽\bar{R_{\theta }}=\frac{1}{N}\sum_{n=1}^{N}\sum_{t=1}^{T_{n}}\left(\sum_{t^{\prime }=t}^{T_{n}}\gamma^{t^{\prime }-t}r_{t^{\prime }}^{n}-b\right)▽logP_{\theta }\left(a_{t}^{n}|s_{t}^{n}\right)$$
where γ is the time decaying factor, b is baseline represented by $V^{\pi }\left(s_{t}^{n}\right)$, $r_{t^{\prime }}^{n}$ represents the reward from time step $t^{\prime }$ to the end of episode. The randomness is introduced due to the omission of expectation E, $Q^{\pi }\left(s_{t}^{n},a_{t}^{n}\right)=E\left[r_{t}^{n}+V^{\pi }\left(s_{t+1}^{n}\right)\right]$. The gradient becomes the following approximation (Equation (Equation 6) [54]) in which sampling methods are used.
(6)$$▽\bar{R_{\theta }}=\frac{1}{N}\sum_{n=1}^{N}\sum_{t=1}^{T_{n}}\left(r_{t}^{n}+V^{\pi }\left(s_{t+1}^{n}\right)-V^{\pi }\left(s_{t}^{n}\right)\right)▽logP_{\theta }\left(a_{t}^{n}|s_{t}^{n}\right)$$

Target network (Figure 3) in our model consists of two networks whose parameters are the same initially. The parameters of target network are fixed to generate the fixed value for the network training with regression, and after training the target network’s parameters are then shared. The sampling work consume much time and we used the replay buffer in which large amount of history samples are stored to counter this challenge.

The DDQN network structure differ from DQN in that the target network is proposed to avoid the problem of moving training object. Initially the parameters of target network $\hat{Q}$ are copied from the current network of Q and then the target network parameters are froze to generate the expected q value $r_{t}+Q^{\pi }\left(s_{t+1},\pi \left(s_{t+1}\right)\right)$. The training object is to minimize the difference between the $Q^{\pi }\left(s_{t},\pi \left(s_{t}\right)\right)$ and $r_{t}+Q^{\pi }\left(s_{t+1},\pi \left(s_{t+1}\right)\right)$ using stochastic gradient descent.

Actor-Critic network structures combine the DDQN (actor) and the state value function (critic). Critic in Q-learning does not directly determine the action, it evaluates the goodness of the actor with the cumulated rewards after visiting states s. When the state s is input to actor $V^{\pi }$, the output scalar $V^{\pi }\left(S_{a}\right)$ represents the value of the state. The main methods of evaluating $V^{\pi }\left(S_{a}\right)$ are the MC and TD approaches. The results differ due to different sampling which MC samples complete trajectories and their returns while TD looks several steps ahead. MC has an unbiased mean and large variance while TD has small variance and biased mean. Importance sampling avails changes from on-policy to off-policy when the reacting actor’s policy is hard to estimate. Agent learned and interacted with environment is the same for on-policy and different for off-policy. Proximal Policy Optimization [18] and Trust Region Policy Optimization [18] ensure the parameters of demonstration policy are not far from the reacting actor in the off-policy scenario. Model-free RL of q-learning uses the target network function to train and approximate the q function. Model based such as Bayesian model-based reinforcement learning or Bayes-adaptive RL. The policy gradient descend is expressed in Equation (Equation 7) [54].
(7)$$\begin{array}{cc} J_{PPO}^{\theta^{\prime }}= & J^{\theta^{\prime }}\left(\theta \right)-\beta KL\left(\theta ,\theta^{\prime }\right)where\\ \; & J^{\theta^{\prime }}\left(\theta \right)=E_{\left(s_{t},a_{t}\right)∼\pi_{\theta^{\prime }}}\left[\frac{p_{\theta }\left(a_{t}|s_{t}\right)}{p_{\theta^{\prime }}\left(a_{t}|s_{t}\right)}A^{\theta^{\prime }}\left(s_{t},a_{t}\right)\right] \end{array}$$
where the *KL* divergence of $\theta ,\theta^{\prime }$ stands for the behavior difference and β is used as the regularizer to control the difference.

The difference between A2C and A3C is that A3C is asynchronous. A3C consists of several independent workers with their own weights who interact with a copy of the environment in parallel. Thus, they can explore a bigger state-action space in much less time in theory. The workers are trained in parallel and update periodically the global at an “asynchronous” time, which holds shared parameters. They explored the environment, calculated individual updates, and sent the updates back to the global. When a worker sends back an update, the global updates itself and then the worker. The workers align their parameters to the global and are synchronized with the freshly updated global after each update. Parameters information flows from the workers to the global and between workers as each worker resets weights according to the global.

We proposed using A3C as our architecture where actor and critic NNs are similar to the generator and discriminator in GAN, respectively. Actor realized the policy function and the critic NN realized the state value function, respectively. The training object is to minimize the summed losses of these two combined functions. Both functions are trained in a single iteration.

Different to existing design of actor and critic using the fully connected NN, the more appropriate NN to function as actor and critic are proposed. For example, CNNs are fit for images tasks and RNNs are for sequential tasks traditionally so the LSTM is considered as the actor. However, the recent NN with attention mechanism achieved remarkable successes and some research [18] even claimed the LSTM should be replaced by attention mechanism. In our work, therefore, we designed TCN with attention mechanism based anomaly detector trained with the actor-critic algorithm of RL to detect the anomalies for sequential tasks. The architectures of NN of the detector could be adapted to the corresponding tasks for better performances. CNNs were employed as the detector for the images related anomalies, and TCN with the attention mechanism was employed for sequential tasks.

### 3.2. Generative Adversarial Learning

There were anomaly detection studies done based on GAN and our proposed model share similarity with GAN [55], the GAN architecture (see Figure 4) for anomaly detection was therefore designed to compare the performances with RL methods. The CNN + LSTM architecture was constructed for the generator and the MLP was for the discriminator. The Generator (G) and Discriminator (D) were trained using the two-player minmax loss game defined as in Equation (Equation 8).
(8)$$L\left(D,G\right)=E_{Z∼p_{z}\left(Z\right)}\left[\log\left(1-D\left(G\left(Z\right)\right)\right)\right]+E_{X∼p_{x}\left(X\right)}\left[logD\left(X\right)\right]$$

*G*’s objective is to learn the distribution of the real data by increasing the error rate of *D* to fool *D* to think that the generated data are real. *G* tries to minimize $E_{z∼p_{z}\left(z\right)}\left[log\left(1-D\left(G\left(z\right)\right)\right)\right]$ while *D* to maximize $E_{x∼∼pr(x)}\left[logD\left(x\right)\right]$.

## 4. Materials and Methods

### 4.1. Definitions

The anomaly detector ‘d’ observed one state $Y_{t}$ at time t and the detecting can be modeled as a partially observable Markov decision process (POMDP) in which not all states can be observed. In our methods the environments were sensed by agents through observational spaces. Anomaly detector d is the agent in RL, the action ‘a’ which d will take under policy π is defined as the probability distribution (Stochastic policy) π:= p(A|S) given the state s, where S and A denote the sets of states and actions, respectively. The policy was modeled as the NN with parameter θ where the inputs were the states and the outputs were the action distribution. In our task, the binary classification of {0, 1} was given in which 1 suggests anomaly and 0 otherwise. Action space is denoted as the possibilities of the actions d will take and then get a reward from. For example, action space in the KDD intrusion environment is {‘Probe’, ‘DoS’, ‘U2R’, ‘L2R’, ‘Normal’}. The goal is to obtain the highest reward in the finite time. At time step t, the selected action (${\hat{a}}_{t}$) together with the ground-truth action ($a_{t}^{*}$) is compared by the reward function to get 1 or 0. If the (${\hat{a}}_{t}$) matches ($a_{t}^{*}$), instant reward ($r_{t}$) of 1 is given, otherwise the reward is 0. The performance of $V_{\pi }$ (Value function in RL) is formalized as:(9)$$V_{\pi }=\sum_{s\in S}w^{\pi }\left(s\right)\sum_{a\in A}R\left(s,a\right)\pi \left(s,a\right)$$
where $w^{\pi }\left(s\right)$ is the probability of the system in the state *s*, *R*(*s*, *a*) represents the cumulated reward starting from state s and action a. The performance is the expected cumulated reward following π. $V_{\pi }$ was realized as the NN in which the current state and action were input to and the network returned the value of the state from the perspective of the whole episode. The optimal performance $\pi^{*}$ satisfies: $\pi^{*}=argmax_{\pi }V_{\pi }$ which means the best performances. π is improved consistently by learning from the experience with the goal of gaining better rewards.

For time series anomaly detection (see Figure 10), the sliding window mechanism ($t_{i}$, $t_{i+1}$, $t_{i+n}$) was used to get the rewards instead of the complete time series. The detector acted according to the policy and was rewarded for every window. After the sliding window swept through the series the rewards were summed up. The red rectangle in the following figure represents the sliding window and the detector acted and got the rewards. If the anomaly was correctly detected judged by the label, the larger rewards (in our tests five point for reward) were obtained and the goal of maximizing the summed rewards then guided the training of the parameters of the policy and value network.

### 4.2. Anomaly Detector Architecture

A2C consists of two independent NNs whose neurons are not shared and are parameterized by θ. A2C improved the value and policy based algorithm by having the critic to learn advantage values which could be interpreted not only the goodness of actions but also room for improvement. A3C proposed by DeepMind [56] further improves A2C in that the former provides the asynchronous functionality and each worker interact with a different copy of the environment in parallel and thus is faster and more flexible. A3C consists of a global agent and several individual workers illustrated in Figure 5. Both global agent and individual learners are modeled as DNNs with each having two outputs: one for the critic and another the actor. The first output is a scalar representing the expected reward of a given state V(s) and the second is a vector representing a probability distribution over all possible actions (s,a).

The proposed A3C framework as the anomaly detector could increase the detection efficiency through utilizing the computing capability of multi-processors. Most of the current work employed basic NNs (for example the feedforward neural network) for the actor and critic structures. We proposed that the design of NNs’ architectures should adapt to the targeted problems. If the anomaly detection of images related area is targeted, the CNN based structure is more appropriate while for sequential task the attention mechanisms is used (see Figure 6 for the two NN architectures). Anomaly detector architecture consists of one global network and several worker agents (12 workers are realized in our test). The cyclic training process involves five steps. (1) Each worker initially reset to global network, (2) interacts with respective environments, (3) estimates value and policy losses, (4) calculates gradient from losses and gradients of all workers are averaged to update the global neural network weights, (5) workers go back to step (1) to carry out another training until convergence or time maximum reached.

Problem definitions such as the environment, system states, action spaces, rewards, states transition, state/value functions, play the vital role in the specific RL modelling task. Generally, problem definitions depend on specific tasks. For the test dataset we defined the environment, states, actions, the dynamics of the states transitions, rewards, and we proposed using appropriate state/action value NN under the A3C framework for different tasks. The algorithm for training anomaly detection based on A3C model was presented as follows.

For each worker neural net samples the batches $s_{t_{1}}a_{t_{1}}s_{t_{1}+1},s_{t_{i}}a_{t_{i}}s_{t_{i}+1}$… from the database and then the states are fed into the policy NN (parameter $\theta^{\prime }$ for worker, θ for global) and state value NN($\theta_{v^{\prime }}$ for worker, $\theta_{v}$ for global). The difference between the estimated value function $R_{t}$ and the actual value function $V_{t}$ (advantage value $A_{t}=R_{t}-V_{t}$) is used to train the NN approximator of policy function. Increase the probability of action $a_{t}^{n}$ if positive advantage is returned and else decrease that probability for the negative advantage. Each worker updates its own $\theta^{\prime }$ and $\theta_{v^{\prime }}$ and these parameters are then submitted to global. The derivative to global parameter dθ, d$\theta_{v}$ are computed on the average of each work $\theta^{\prime }$, $\theta_{v^{\prime }}$ to prevent the overestimate of some actions. The entropy of policy $(H\left(\pi \left(a_{j}|\;s_{j};\theta^{\prime }\right)\right)$ is added to encourage exploration and avoid early suboptimal convergence.

The neural sampler provides batches of training samples $s_{t_{1}}a_{t_{1}}s_{t_{1}+1},s_{t_{i}}a_{t_{i}}s_{t_{i}+1}$… (see Figure 7 for the training process). The detector executes the action at under state $S_{t}$ and gets current reward $r_{t}$ and the next time state $s_{t+1}$ from the environment. The policy loss function is in the form of $A_{t}\log\pi \left({\hat{a}}_{t}\right)$, the log probability of the selected action ${\hat{a}}_{t}$ under the probability distribution and weighted by the corresponding advantage value $A_{t}$. The actor and critic NN are constructed and trained simultaneously by minimizing both value and policy loss functions using the stochastic gradient descent (SGD) method (see Algorithm 1). For prediction we use the NN that implements the policy function by selecting the action with the highest probability. We summarized some common elements and differences among the RL, GAN and the proposed models in Table 2.
**Algorithm 1** Algorithm for anomaly detection based on A3C.  1:Initialize global shared parameters θ, $\theta_{v}$ and counter N:=0;  2:Initialize worker thread parameters $\theta^{\prime }$, $\theta_{v}^{\prime }$ and time step t:=1;  3:**while** N < $N_{max}$
**do**  4:   dθ:=0, d$\theta_{v}$:=0;  5:   $\theta^{\prime }$:=θ, $\theta_{v}^{\prime }$:=$\theta_{v}$;  6:   $t_{start}$:=t, get state $s_{t}$;  7:   **while** ($s_{t}$ is NOT terminal) or (t- $t_{start}$<=$t_{max}$) **do**  8:      perform actions $a_{t}$ = $\pi (a_{t}|s_{t};\theta^{\prime })$;  9:      get reward $r_{t}$ and transition to $s_{t+1}$;10:      t:=t+1, N:=N+111:   **end while**12:   **if** ($s_{t}$ is terminal state) **then** R:=0; R:=V($s_{t}$, $\theta_{v}^{\prime }$);13:   **end if**14:   **for**
$j\in (t-1,\ldots t_{start})$
**do**15:      R:=$r_{j}$+γR;16:$$\begin{array}{cc} \;\;\;\;\;\;\;\;\;\;\;d\theta :=d\theta & +\partial log\pi \left(a_{j}|s_{j};\theta^{\prime }\right)\left(R-V\left(s_{j};\theta_{v}^{\prime }\right)\right)/\partial \theta^{\prime }\\ & +\beta \partial H\left(\pi \left(a_{j}|s_{j};\theta^{\prime }\right)\right)/\partial \theta^{\prime }; \end{array}$$17:$$\;\;\;\;\;\;\;\;\;\;\;d\theta_{v}:=d\theta_{v}+\partial {\left(R-V\left(s_{j};\theta_{v}^{\prime }\right)\right)}^{2}/\partial \theta_{v}^{\prime }$$18:   **end for**;19:   update dθ and d$\theta_{v}$ to global asynchronously20:**end while**

We proposed replacing the original MLP with CNN for the anomaly detection in images/video areas and using attention mechanism for sequential tasks. The inputs to the CNN network are three dimensional tensors representing the images or the frames of the video. The network structure has three convolution layers which convolves the inputs in order with some fixed number of filters and strides which are dependent on the specification of tasks. The last layer is fully-connected hidden layer of fixed number of units projects to the Q-value. All these layers are separated by activation Rectifier Linear Units (ReLu). The optimizer of “RMSProp” with momentum is employed to train the network. Using attention mechanism instead of LSTM for sequential task is to enhance the efficiency while retaining the effectiveness.

## 5. Results

We conducted evaluation tests on three benchmark datasets (AWID, NSL-KDD, Time Series Dataset) and compared with other methods to demonstrate the effectiveness of our method. We also tested the latest dataset of [57] ’CIC-IDS-2017’ (https://www.unb.ca/cic/datasets/ids-2017.html, accessed on 1 February 2021) and [58] ’CIRA-CIC-DoHBrw-2020’ (https://www.unb.ca/cic/datasets/dohbrw-2020.html, accessed on 1 February 2021) from Canadian Institute for Cybersecurity. The number of samples for DDoS was 225,745 with 78 attributes and 1 redundant attribute was deleted during preprocessing. NSL-KDD dataset was although old and there were even research papers arguing against using the dataset, however this dataset was the benchmark or standard for network traffic detection which led to the development of network intrusion detection. Furthermore, the issue of low detection rate for U2R and R2L remained. The detection rate for these two anomalies were very low even for the NSL-KDD cup winner back then. There were few conference papers reporting the detection rate for the two anomalies was improved greatly. However, the realization details were missing and could not be replayed. NSL-KDD and several different new datasets including AWID and real-world time series were introduced in the experiments to address the deficiency.

Metrics of accuracy, precision, recall and F1 were defined in Equation (Equation 10). P, N, TP, FP, TN, FN stand for positive, negative, true positive, false positive, true negative, and false negative, respectively. Precision (also Positive Predictive Value) is the fraction of relevant instances among the retrieved instances, while recall (or sensitivity) is the fraction of the total amount of relevant instances that were actually retrieved.
(10)$$\begin{array}{cc} Accuracy & =\frac{TP+TN}{TP+FP+TN+FN}\\ Precision & =\frac{TP}{TP+FP}\\ Recall & =\frac{TP}{TP+FN}\\ F1 & =\frac{2\times Precision\times Recall}{Precision+Recall} \end{array}$$

### 5.1. AWID Datasets Test

AWID datasets were focused on intrusion detection for wireless networks (http://icsdweb.aegean.gr/awid/index.html, accessed on 1 February 2021). We used the reduced “AWID-CLS-R-Trn” and “AWID-CLS-R-Tst” as the training and test datasets, respectively. Original record includes 154 attributes and 1 class attribute which denotes the record is normal or under attack. The complete features should be analyzed to extract useful ones from noises which could promote prediction capability and reduce complexity. We preprocessed the complete features set into 46 features and 1 class types excluding the invalid and misleading features. Different attacks are organized into a normal and three types, namely, flooding, impersonation and injection. The number of four kinds of samples are listed in Table 3.

Ten-fold cross validation method was used for the training and testing. The dataset was reshuffled randomly to avoid sample’s similarity in each group and was divided into 10 groups with 9 of them for training and the left one group for testing. The process repeats 10 times. During training the Keras early stopping callback was used to avoid overfitting. During training five optimizers (adam, rmsprop, adagrad, adadelta, and sgd) were tested. The best performed optimizer of “Adam”, categorical cross-entropy loss function for multi-classification and $10^{-4}$ for the learning rate were chosen. MLP design was straightforward with three fully connected layers, dropping out set, optimizer of “adam”, loss function of MSE, batch size of 100 and around ten thousand (early dropping) weights needed to be back propogated to reach the minimum loss rate.

We compared and listed results in Table 4 the detection performances using the proposed model with MLP, SMOTE and adversarial RL models.

### 5.2. Time Series Anomaly Test

Sequential anomaly detecting tasks are faced with learning from experience on the fly. Rewards and discounts in theory are part of the problem definitions but in practice they are well-designed parameters and can be tweaked. Two dataset are used for this test. The first contains 367 Yahoo time series with each labeled as normal or not and these are used as environment for training to establish the anomaly detector. The action space is the {anomaly, normal} taken by the detector. The test set consists of 58 simulated and real-world time series (such as traffic, exchange and tweets) with anomalies.

The sliding window mechanism was used to detect the anomalies in the given time series. The size of the slide window for state and reward functions were set to 25, reward was 5 points for true positive, −1 point for false positive or error alarm, 1 point for the true negative and −5 for the false negative or miss alarm. Red and green 3-D plot in Figure 8 represent different action-value when the detector adopted different actions. The discount factor, epsilon for exploration and the episode were set to 0.8, 0.1 and 1000, respectively. The test results showed loss rate for the two different actions were below 3%.

We compared the performances of the proposed anomaly detector with the time series anomaly detector using RNN and Q-learning in RL. The results for the second dataset were demonstrated in Figure 9 where red and blue curve(a) represent the real and forecasted value for the training and testing. The blue-shadowed area around 4500 indicated anomalies and the curve(b) showed smoothed error between real and forecasted values. The best test score using the RNN and Q-learning of RL method was precision = 52.3%, recall = 100% and F1 = 68.9% among 58 labeled real-world and artificial timeseries data files and it was demonstrated in Figure 10. The precision rate for all the test set was unsatisfactorily low. Same test set was run on the proposed anomaly detector and the best score was precision = 75.3%, recall = 95.8% and F1 = 84.3%. Blue, green and red curve in Figure 10 represent the aforementioned targeted time series with best scores, action and rewards based on the actions, respectively, for the test result.

### 5.3. NSL-KDD Network Anomaly Test

This dataset organize different attacks into four types, namely, DoS, Probe, R2L and U2R. For example, ‘neptune’, ‘teardrop’, ‘land’, etc. are grouped in the DoS attack, ‘portsweep’ and ‘nnmap’, etc. the Probe, the R2L, and ‘guess_passwd’ and ‘named’, etc. ‘buffer_overflow’, ‘rootkit’, etc. the U2R.

Generally dataset needs to be preprocessed first for follow-up RL modelling process and exploratory data analysis is used to get some statistics(such as the number of features and data), data cleaning to delete those invalid features, the standardization of the numeric features and the visualization of the high-dimensional data. We used the t-SNE approach on the NSL-KDD training dataset which consists of 125,974 data points with 122 features each and classified as either 1 or 0. The t-SNE illustration in Figure 11 showed the relatively clear boundary (red stands for 0, black 1) of the binary classification. Nine traditional methods of anomaly detection were compared using the samples of NSL-KDD training data with dimension reduction. The red circles in Figure 12 showed the not so clear decision boundary for each detector.

The datasets are used as the environments in the RL tasks and they are split as the training and test. There are 122 features and 5 classes (4 anomaly and 1 normal) for the specific NSL-KDD dataset and the 122 observational states are used as the system states. Four anomaly (Probe, DoS, U2R, L2R) and one normal classification are defined as the action space. The correctly recognized anomalies are rewarded 3 points and −3 points otherwise.

Machine learning algorithms such as Random forest (RF), Bayes, adaboost, etc. and our proposed method all achieved around 99 percent of precision for anomaly of ‘DDoS’, ‘Portscan’. For imbalanced datasets (much fewer negative samples) of ‘Bot’, ‘web attack’ and ‘infiltration’ the precision of benign traffic was around 99 percent, the precisions of three different web attacks (Brutal force, XSS, SQL injection) were around 70%, 45% and 99% with RF. Our proposed method achieved same precision for benign traffic and slightly better accuracies for three web traffics. Due to page size limit we did not display all the results data. Results of classification using RF was displayed in Figure 13. Our proposed methods also achieved nearly the same precision of 99%.

Detection rate for Probe and Dos using KNN, C4.5 were acceptable while anomaly of U2R and R2L were unsatisfactorily low as demonstrated in the confusion matrix (see Figure 14b) where U2R was often misclassified as Probe and R2L as normal. The test scores were low for attacks such as “warezmaster”, “saint” and “processtable” in Figure 14a. Although the detection rate of these anomalies using RL has already increased to around 20 percent which surpassed the results from KDD-CUP winner (13.2% for U2R, 8.4% for R2L in [59,60]), there is still large space for improvement. The relative low number of training samples and these anomalies embedding the data in the payload of normal or probe traffic might lead to the poor performances of the anomaly detector. There were papers [61,62] reporting the detection rate was improved greatly, however the realization details to support the conclusions was not very clear. Even if the conclusion of improved performances was confirmed, the potential overfitting that might lead to the high detection rate and the generalization ability to other tasks still remained low.

The statistical one sample sign test was used for the results. Each method repeat 10 times (using same parameters and random seed number) and we checked the raw data to avoid the overfitting errors and used the average to represent the finals. The averages became the null hypothesis. We computed the obtained frequency which was above the average, and the expected frequency was 5, which meant five below average and five above. Chi-square value, degrees of freedom, and alpha level of 0.05 was set to determine the hypothesis should be rejected or not. The average hypothesis were accepted and the ranges were ±0.5% for the metrics of accuracy, precision, recall.

Comparison of the rewards and losses of the proposed model with other RL methods was illustrated in Figure 15. Blue curve with circle represents the proposed model which surpassed the existing RL models in terms of rewards and loss rates. The training time for the RL models was around 25 s per epoch. We tested 500 epochs and found after 100 epochs the performances were not improved (we considered after 100 epochs the model converged). So the average time for training one model was around 2500 s. We compared the training time for the whole 300 epochs between the proposed model and others (see Figure 16). The detection time (test time) was about 15 s for 22,544 samples. Results of accuracy, precision, recall rate and F1 were listed in Table 5 where numbers in bold represent the best score for the specific metric. The proposed model outperformed other methods in terms of accuracy, precision and F1 score.

Inspired by [55], we compared anomaly detection performances between GAN and the proposed model based on (1) actor and critic NNs function similarly to the generative and adversarial NNs of GAN, respectively, (2) these two employed same methods of ‘freezing learning’, ‘batch normalization’. In proposed RL models the discount γ was set to 0.95, learning rate α=0.002 and the exploration rate set to 0.85. The number of steps between target network updates was τ = 1000. Overall training is 10 M steps. The agent is evaluated every 100 K steps, and the best policy across these evaluations is kept as the output of the learning process. The size of the experience replay memory is 100 K tuples. The memory gets sampled to update the network every four steps with minibatches of size 32. The simple exploration policy used is an ε-greedy policy with the decreasing linearly from 1 to 0.1 over 100 K steps.

## 6. Discussion

The proposed adaptable RL architecture of A3C framework employs appropriate policy and value DNNs for specific domains which are TCN structure with attention for sequence and CNN for images or video. In theory, for traditional RNN and its variant LSTM sequence models, hidden state matrix is $O(d^{2})$ where d is the dimension and input n is often the sequence length or sliding window size. The time complexity is $O(n\times \left(d^{2}\right))$ where typically d = 1000 and n is in the order of less than 100. The sliding window we used in tests is 25. Hence the complexity is $O(25\times 1000^{2})$. The complexity is determined by input and hidden state matrix. In contrast, the proposed TCN with attention, the complexity is $O(\left(n^{2}\right)\times d)$. This is $O(25^{2}\times 1000)$. The input length is far less than hidden state which might explain why the TCN with attention is much time efficient than the traditional counterparts. In practice, we compared the performances between LSTM and attention mechanisms and found that the training time reduced to around 80 percent by the attention model while the accuracy remained comparable [63]. That was one reason TCN with attention instead of LSTM for anomaly detection in this paper.

Previous anomaly detection approaches were challenged by many problems especially the data labeling and sequence prediction. Applying RL provides a new angle to the anomaly detection tasks and the test results indicated the effectiveness of this method. Actor-critic was a promising architecture in that actor could select actions with the least computation especially for continuous actions and the stochastic policy could be learned for the best probability of all kinds of actions.

Different to large labeled datasets like ImageNet accelerated the development of supervised learning, lack of standardization of environments(equivalent to the labeled datasets) in RL might hinder the application of RL to the anomaly field which also makes it difficult to reproduce published research and compare results with.

## 7. Conclusions

The proposed anomaly detection model consists of one global and twelve workers which was provided with a copy of the environment, trained independently and then updated the global asynchronously. An attention mechanism was proposed for the actor network of global and each worker in detecting sequential anomalies. CNN architecture was for the task of detecting images, video anomalies. The critic was constructed with the Feed Forward NN with three layers connected with ReLU activation function. Dynamic learning rate was used to update weights of the actor under the guidance of critic. Larger learning rate was set initially and decayed over time to make the training process fast and stable.

Test results based on three benchmark datasets showed our model’s effectiveness in detecting anomalies and the performances surpassed many existing counterparts or at least comparable to these models.

## Figures and Tables

**Figure 1 entropy-23-00274-f001:**
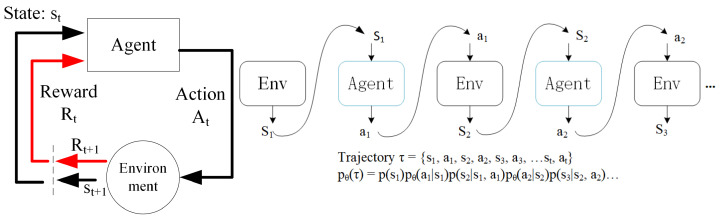
Agent environment interaction (**left**), action and reward sequences (**right**).

**Figure 2 entropy-23-00274-f002:**
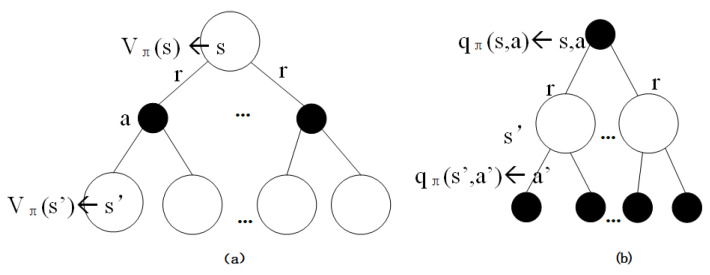
(**a**) Illustration of state value computation. (**b**) Action value.

**Figure 3 entropy-23-00274-f003:**
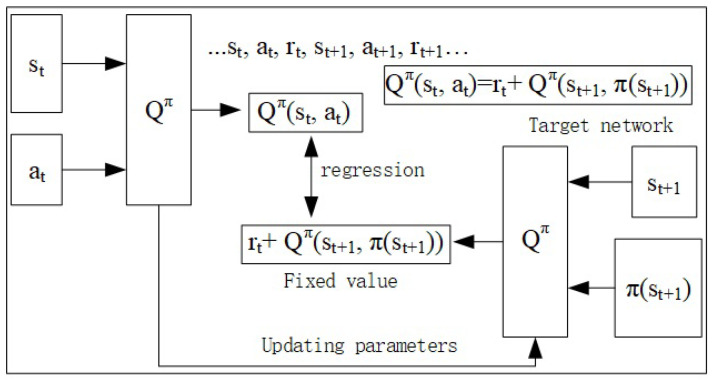
Target network illustration.

**Figure 4 entropy-23-00274-f004:**
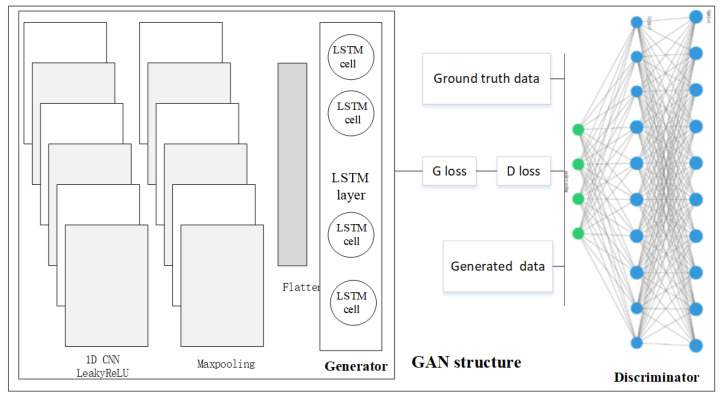
Anomaly detection architecture based on GAN.

**Figure 5 entropy-23-00274-f005:**
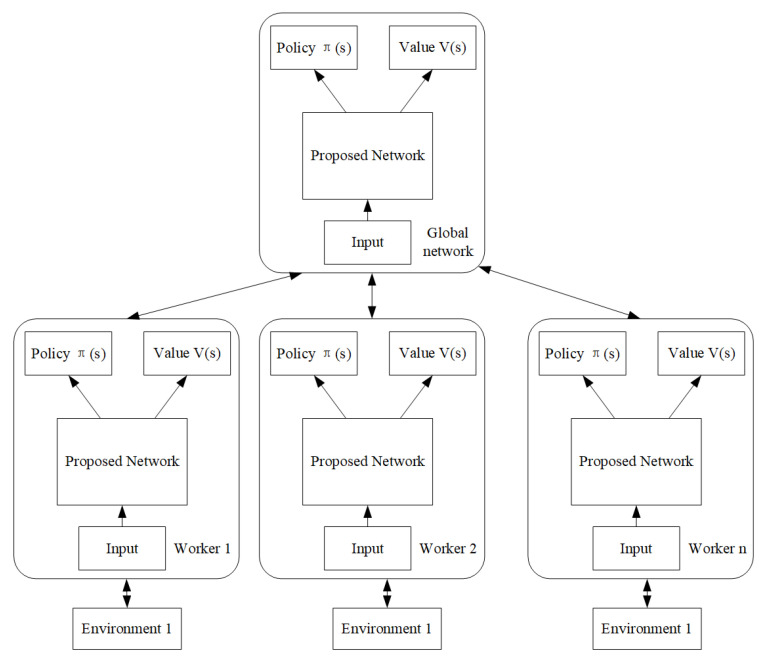
Anomaly detection architecture based on A3C.

**Figure 6 entropy-23-00274-f006:**
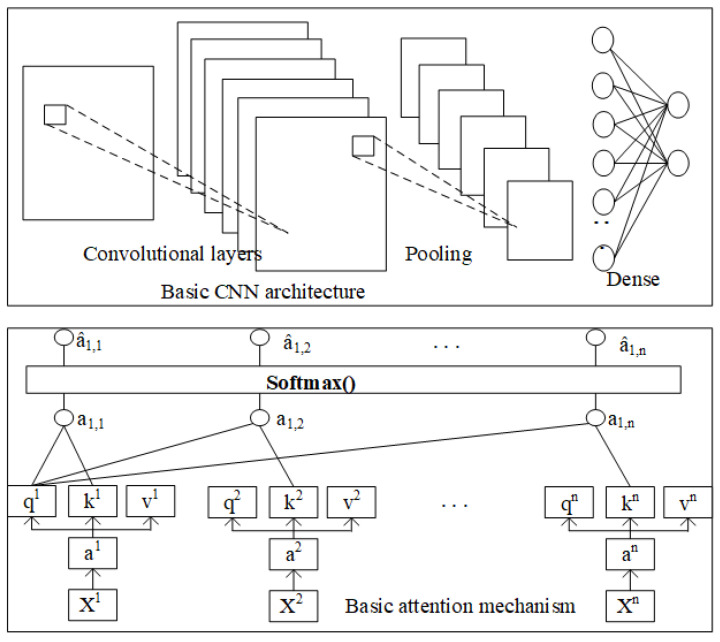
Proposed network for actor.

**Figure 7 entropy-23-00274-f007:**
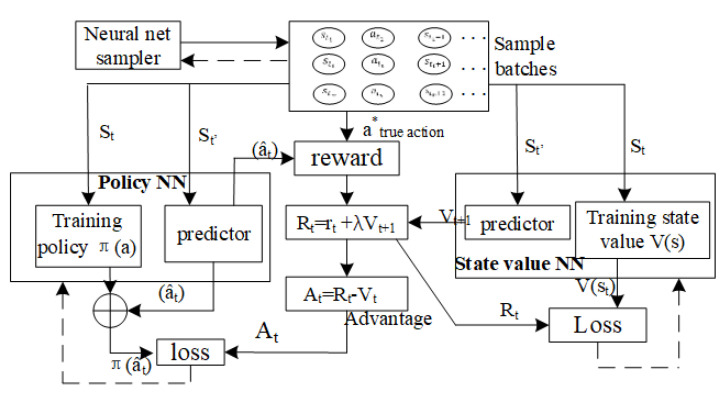
Training process for detector and evaluator.

**Figure 8 entropy-23-00274-f008:**
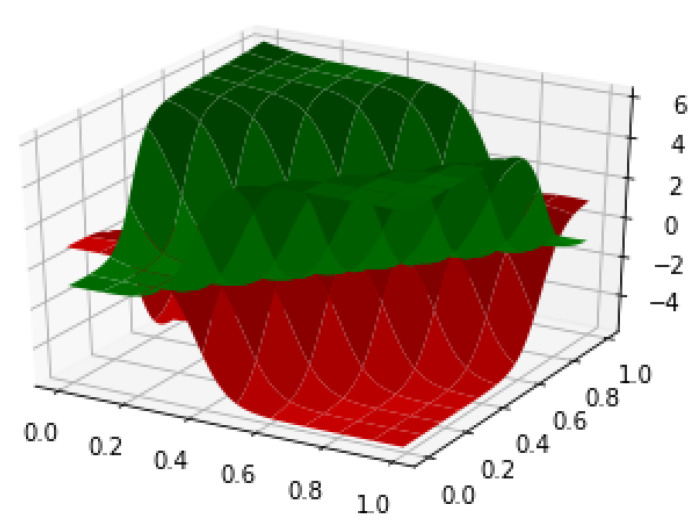
Action-value function for actions of time series anomaly.

**Figure 9 entropy-23-00274-f009:**
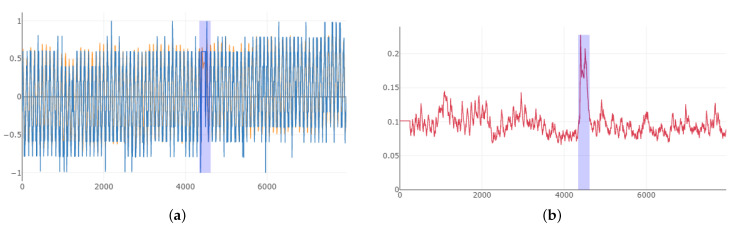
(**a**) Time series anomaly test results without smooth. (**b**)smoothed error for the time series anomalies.

**Figure 10 entropy-23-00274-f010:**
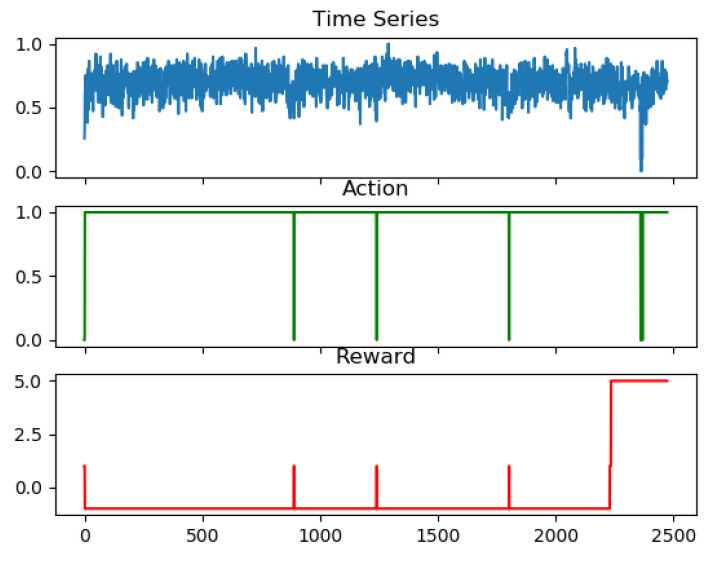
Time series anomaly using the RNN and Q-learning of RL.

**Figure 11 entropy-23-00274-f011:**
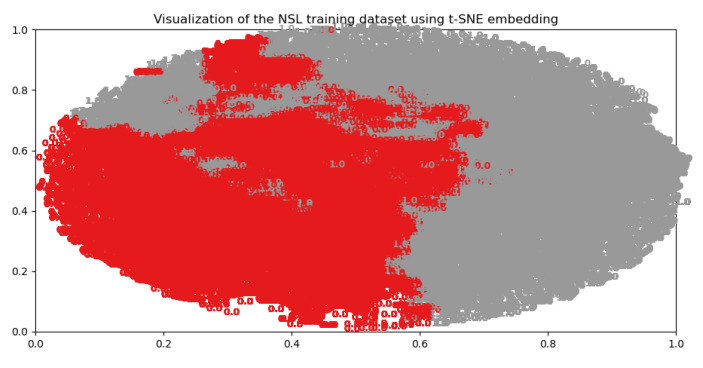
t-SNE illustration using NSL-KDD dataset.

**Figure 12 entropy-23-00274-f012:**
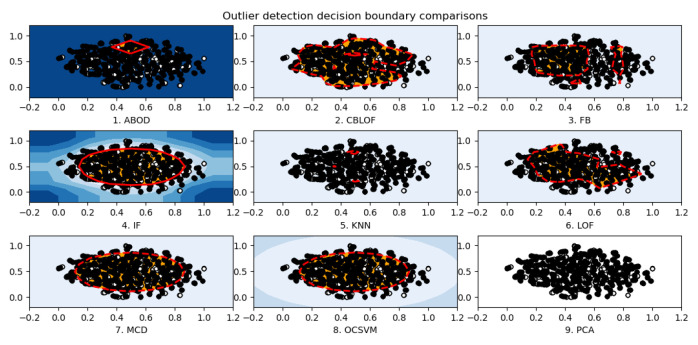
Outlier decision boundary comparisons using NSL-KDD dataset.

**Figure 13 entropy-23-00274-f013:**
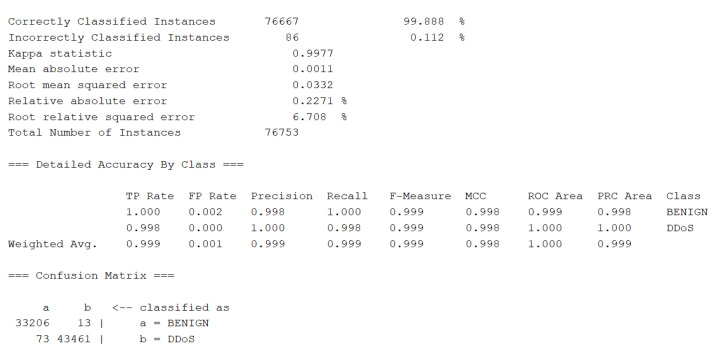
Detailed results for dataset CIC-IDS-2017 using RF algorithm.

**Figure 14 entropy-23-00274-f014:**
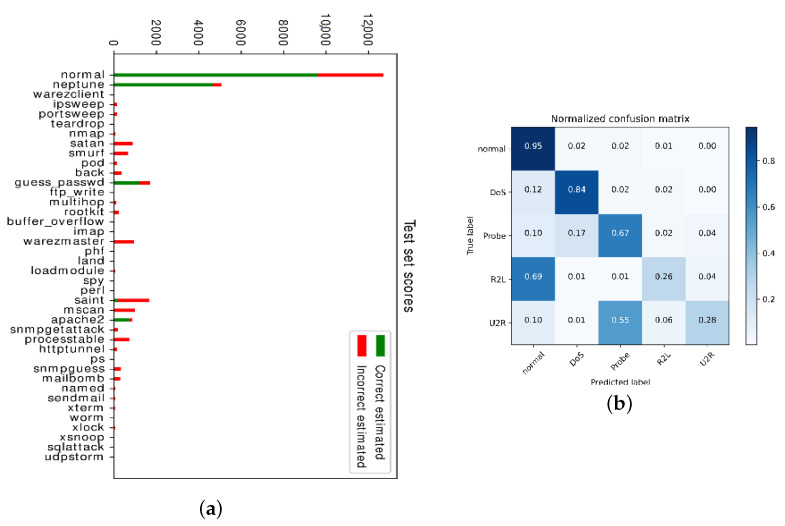
(**a**) Test scores for different attacks. (**b**) Confusion matrix for five anomalies.

**Figure 15 entropy-23-00274-f015:**
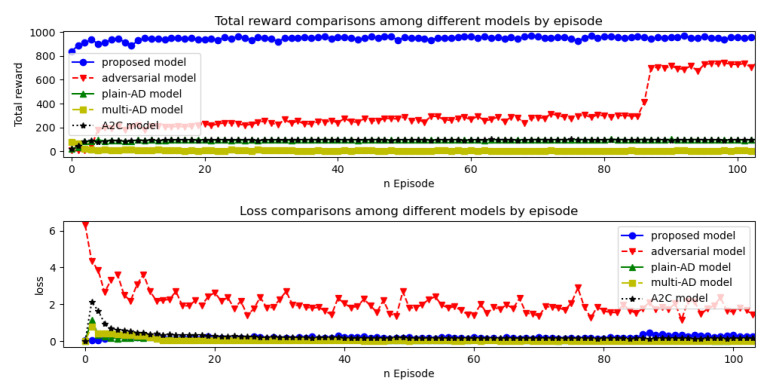
Rewards and losses comparisons among different models by epochs.

**Figure 16 entropy-23-00274-f016:**
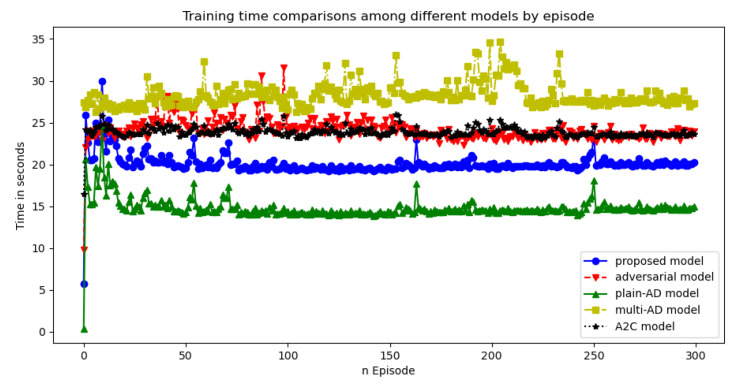
Training time comparisons among different models by epochs.

**Table 2 entropy-23-00274-t002:** Comparaions among RL, GAN and the proposed models.

Methods	Algorithms	Application	Features
Value-based	Q-learning, DDQN, etc.	Simple environment	Sample efficient, steady
Policy-based	Policy gradient, REINFORCEMENT	continuous, stochastic	Unstable, time consuming
		environment	
Actor-Critic (AC)	Combine Value and policy, Actor(value) computing actions, Critic (policy) producing q value of actions;outperformed Value, Policy separately	Complex environment (such as 2/3D video gaming)	Batch normalization Target network Replay buffer Entropy regularization Compatibility, etc.
GAN	Generator, Discriminator	Image video/audio, etc.	Batch normalization, label smoothing, etc.
A2C	Introducing advantage based on AC, evaluate goodness of action and its improvement space	Complex environment	reduces high variance of policy networks, stable
A3C	One global, several workers share the same environment; Introducing asynchronous based on A2C	Complex environment	further improve efficiency based on A2C, robustness, speed
Proposed method	Similar to A3C	Complex environment	Adaptable, Image: CNN Sequence: TCN

**Table 3 entropy-23-00274-t003:** Numbers for AWID reduced training and test dataset.

AWID-CLS-R-Trn		AWID-CLS-R-Tst	
Flooding	48,484	Flooding	8097
Impersonation	48,522	Impersonation	20,079
Injection	65,379	Injection	16,682
Normal	1,633,190	Normal	530,785

**Table 4 entropy-23-00274-t004:** Test results comparisons among proposed method and others.

Method	Type	Accuracy (±0.5%)	Precision (±0.5%)	Recall (±0.5%)	F1 (±0.5%)
MLP	Flooding	0.9488	0.9195	0.6288	0.7469
	Impersonation	0.9326	**0.6195**	0.5491	0.5822
	Injection	0.9496	0.9202	**1.00**	0.9584
	Normal	0.9872	0.9498	**1.00**	0.9732
Adversarial RL model	Flooding	**0.9941**	0.9452	0.6183	0.7476
	Impersonation	0.9479	0.3201	0.4392	0.3703
	Injection	**0.9980**	0.9358	0.9999	**0.9668**
	Normal	0.9400	0.9727	0.9620	0.9673
SMOTE	Flooding	0.9924	0.6001	0.6211	0.6104
	Impersonation	0.9521	0.3398	0.9303	0.4978
	Injection	0.9720	0.4105	**1.00**	0.5821
	Normal	0.9396	0.9897	0.8769	0.9299
Proposed Anomaly Detector	Flooding	0.9930	**0.9847**	**0.6430**	**0.7780**
	Impersonation	**0.9571**	**0.4736**	**0.9402**	**0.6299**
	Injection	0.9834	**0.9848**	0.8679	0.9227
	Normal	**0.9917**	**0.9903**	0.9851	**0.9877**

**Table 5 entropy-23-00274-t005:** Metrics comparisons among proposed model and others.

Method	Algorithm	Accuracy (±0.5%)	Precision (±0.5%)	Recall (±0.5%)	F1 (±0.5%)
Linear model	OCSVM	0.6542	0.6953	0.6512	0.6725
Ensemble	IF	0.7911	0.8483	0.7192	0.7784
Proximity	LOF	0.6834	0.7923	0.6597	0.7189
	KNN	0.7808	0.8933	0.6706	0.7661
NN methods	VAE	0.7912	0.8311	0.7018	0.7610
	MLP	0.7966	0.8679	0.6647	0.7529
GAN	WGAN	0.6149	0.8279	0.665	0.7376
RL methods	Plain-AD	0.7630	0.8902	0.7829	0.8331
	Multi-AD	0.7265	0.8882	0.7265	0.7987
	A2C	0.7905	0.8930	**0.7905**	0.8386
	Proposed Anomaly Detector	**0.7920**	**0.9110**	0.7901	**0.8463**

## Data Availability

Not applicable.

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
