# Peer review of "Application of Improved Asynchronous Advantage Actor Critic Reinforcement Learning Model on Anomaly Detection"

_entropy, 2021, doi:10.3390/e23030274_

Round 1

Reviewer 1 Report

This paper aimed at proposing an adaptable asynchronous advantage actor-critic model of reinforcement learning to this field. The performances were evaluated and compared among classical machine learning and the generative adversarial model with variants. Basic principles of the related models were introduced firstly. The problem definitions, modelling processes and testing were detailed. The proposed model differentiated the sequence and image from other anomalies by proposing appropriate neural networks of attention mechanism and convolutional network for the two kinds of anomalies respectively. The paper has been written and organized well. The problem statement, motivation and contribution has been clearly stated.

The comments are given below:

  • It would be good if the citation could be included in Table 1.
  • In section 2, it would be interesting if author can divide the discussion into section subheadings, ML for anomaly detection, DL for anomaly detection. Author has cited a number of good articles however, the discussion one ach citation is very limited. Authors should provide a brief discussion on each citation, its contribution, limitations etc,.
  • Give citations to the equations because similar equations re already used by other researchers.
  • Please correct the Table numbers.
  • Please label the Figure 8, 11, 12

Reviewer 2 Report

This paper proposing an adaptable asyn chronous advantage actor-critic model of reinforcement learning to this field. The performances were evaluated and compared among classical machine learning and the generative adversarial model with variants.

I would recommend to compare the current results with more recent benchmark and algorithms such as transformers, deep learning approaches such as 1D-CNN, 2D-CNN, LSTM, BiLSTM and traditional machine learning classifiers such as SVM, logistic regression and etc

I suggested to add the training accuracy and training time for each model.

The parameters for each model such as MLP should be added.

The Fig 3 does not look good, Fig 3 can be shown as form of Table.

I would suggest to update the related work with following paper:

Yu, Z., Machado, P., Zahid, A., Abdulghani, A.M., Dashtipour, K., Heidari, H., Imran, M.A. and Abbasi, Q.H., 2020. Energy and performance trade-off optimization in heterogeneous computing via reinforcement learning. Electronics9(11), p.1812.

Author Response

This manuscript is a resubmission of an earlier submission. The following is a list of the peer review reports and author responses from that submission.

Round 1

Reviewer 1 Report

This paper suggests an improved asynchronous advantage actor critic reinforcement learning model applied for anomaly detection.

The state of the art, the different methods and tools used, as well as the conducted experiment are excellently presented and argued.

In fact, I have rarely reviewed an article of such high quality on all levels.

I tried to find a weakness in the different parts dealt with in the article, but to no avail.

Here are some observations to better clarify some points of the article:

- Expression (4) is not an equation.

- In Algorithm 1, the use of t and T is confusing, more explanation is desirable.

- In section 5, details regarding the structure of the neural networks used are recommended. In addition, the methods used to optimize the chosen attributes must be added.

Reviewer 2 Report

The paper aims "at proposing an adaptable asynchronous advantage actor-critic model based on reinforcement learning"... is not a3c an RL algorithm, why is it "based" on RL? Not sure what was that, but anyway, here are things which I could not really understand from this study:

1. Anomaly detection as a rule means unsupervised learning, e.g. no ground truth given, in the algorithm proposed, correct labels are fed to both policy and value networks via the reward function, which means the algorithm should not really differ from standard supervised binary classifier built from either attention or cnn (depending on the task) input layers and dense hidden layers 

2. I was not able to understand how the algorithm presented differed from standard a3c, e.g. from figures 8 and algorithm 1, it looks just like standard actor-critic setup

3. When you apply RL for intrusion detection like this, does not every state become terminal, i.e. we label a sample with either 0 or 1, and that's it, the game is over, or does it have something to do with how the samples are distributed in time in the dataset? In this case, does not the RL algorithm just "learn" the dataset instead of features, how would you even apply the model trained this way in a real-world scenario? 

4. In the section with results, accuracy of MLP classifier is suspiciously low. Did you use a validation set and "early stopping" to avoid overfitting? How did you deal with the sample distribution being biased? How many layers / neurons did MLP have? 

5. In Section 3, so much space is spent to describe ddqn, even though a2c is used in the end. There almost no information on how does a2c work. If I had not known it in advance, I would've not learnt it from this manuscript.

6. Figures should have the same or at least similar style. Quality of the figures can also be improved, and vector graphics should be used if possible. Some figures are also not necessary, e.g. that figure with tp / fp / tn / fn square, or figures with CNN structure or LSTM cell. 

7. One more thing about asynchronous learning, it does not look like it really gives any advantages over multiple copies of standard a2c: https://openai.com/blog/baselines-acktr-a2c/  

8. The last but not the least thing the authors should address is the language, since some parts of he manuscript are really hard to read, which makes a reviewer angry and probability of the manuscript to be accepted lower :) 

Round 2

Reviewer 2 Report

Authors made an effort to address minor issues, but the key ones still remain:

  1. The problem considered in the manuscript is simply classification, not anomaly detection
  2. It is standard (not improved) actor-critic which is for some reason used for classification
  3. Experiments with datasets of type  KDD, CICIDS17, etc do not really make sense in this setup
  4. Comparing a neural network with attention layers for feature extraction with basic MLP is not a fair comparison